# SENTINEL: Taming Uncertainty
# with Ensemble based Distributional Reinforcement Learning

**Hannes Eriksson**[1, 2]          **Debabrota Basu**[3, 4]          **Mina Alibeigi**[1]          **Christos Dimitrakakis**[2,5]

[1]Zenseact AB, Gothenburg, Sweden
[2]Chalmers University of Technology, Gothenburg, Sweden
[3]Scool, INRIA Lille-Nord Europe, Lille, France
[4]CRIStAL, CNRS, Lille, France
[5]University of Neuchatel, Switzerland and University of Oslo, Norway

## Abstract

In this paper, we consider risk-sensitive sequential decision-making in Reinforcement Learning (RL). Our contributions are two-fold. First, we introduce a novel and coherent quantification of risk, namely composite risk, which quantifies the joint effect of aleatory and epistemic risk during the learning process. Existing works considered either aleatory or epistemic risk individually, or as an additive combination. We prove that the additive formulation is a particular case of the composite risk when the epistemic risk measure is replaced with expectation. Thus, the composite risk is more sensitive to both aleatory and epistemic uncertainty than the individual and additive formulations. We also propose an algorithm, SENTINEL-K, based on ensemble bootstrapping and distributional RL for representing epistemic and aleatory uncertainty respectively. The ensemble of K learners uses Follow The Regularised Leader (FTRL) to aggregate the return distributions and obtain the composite risk. We experimentally verify that SENTINEL-K estimates the return distribution better, and while used with composite risk estimates, demonstrates higher risk-sensitive performance than state-of-the-art risk-sensitive and distributional RL algorithms.

## 1 INTRODUCTION

Reinforcement Learning (RL) algorithms, with their recent success in games and simulated environments [Mnih et al., 2015], have drawn interest for real-world and industrial applications [Pan et al., 2017, Mahmood et al., 2018]. In addition, since in RL the environment is by definition unknown to the agent, exploring it so as to improve performance and eventually obtain the optimal policy entails risks. Although the risk is not an issue in simulation, it is important to consider risks when interacting in the real world [Pinto et al., 2017, García and Fernández, 2015, Prashanth and Fu, 2018]. In this paper, we employ a model-free approach that enables us both to efficient in terms of the amount of data needed, and to be flexible with respect to the risk metric the agent should consider when making decisions.

Risk sensitivity in reinforcement learning and Markov Decision Processes (MDPs) has sometimes been considered under a minimax formulation over plausible MDPs [Satia, 1973, Heger, 1994, Tamar et al., 2014]. Alternative approaches include maximising a risk-sensitive statistic instead of the expected return [Chow and Ghavamzadeh, 2014, Tamar et al., 2015, Clements et al., 2019]. In this paper, we focus on the second approach due to its flexibility. Either approach requires estimating the uncertainty associated with the decision-making procedure. This uncertainty includes both the inherent randomness in the model and the uncertainty due to imperfect information about the true model. These two type of uncertainties are called *aleatory* and *epistemic* uncertainty respectively [Der Kiureghian and Ditlevsen, 2009].

In recent literature, researchers have either quantified epistemic and aleatory risks separately [Mihatsch and Neuneier, 2002, Eriksson and Dimitrakakis, 2020] or considered an additive risk formulation where their weighted sum is minimised by an RL algorithm [Clements et al., 2019].

In this work, we propose a *composite risk* formulation in order to accurately capture the combined effect of aleatory and epistemic uncertainty for decision-making in RL (Section 4). Our composition of risks relies on *coherent* risk measures, for which we show that their composition remains coherent. Our choice of focusing on coherent risk measures is also motivated by its extensive use and corresponding benefits in control theory Majumdar et al. [2017], decision theory Pflug and Pichler [2016], and reinforcement learning theory [Tamar et al., 2016, Ruszczyński, 2010, and references therein].

We incorporate composite risk measures within the Distribu-

*Accepted for the 38th Conference on Uncertainty in Artificial Intelligence* (UAI 2022).

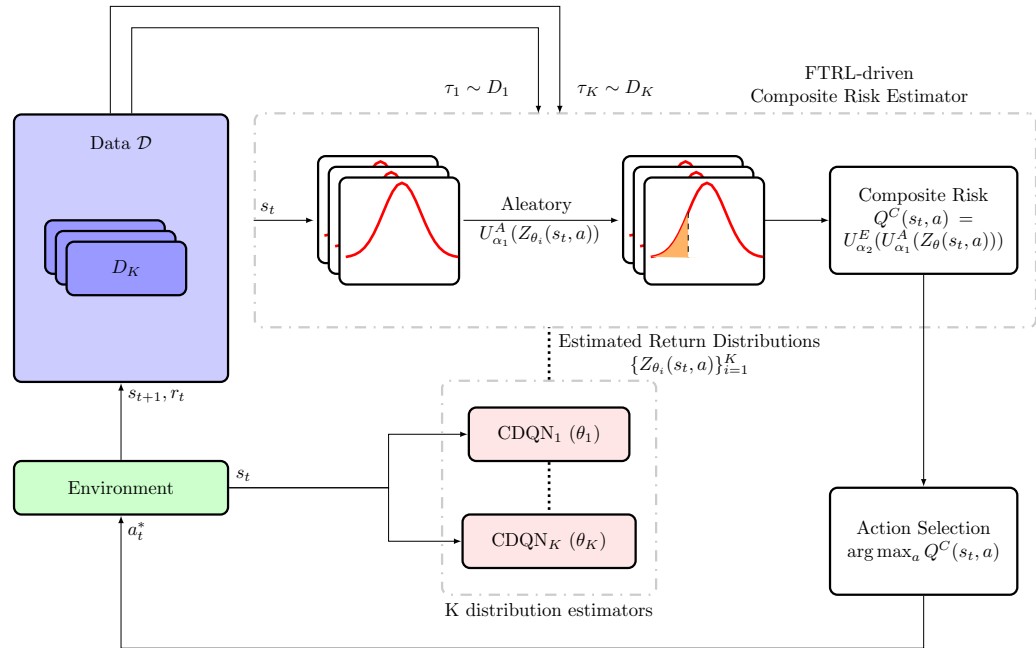

Figure 1: SENTINEL-K with FTRL-driven composite risk estimator and K CDQNs as return distribution estimators.

tional RL (DRL) framework [Bellemare et al., 2017, Tang and Agrawal, 2018, Rowland et al., 2019]. The DRL framework aims to model the distribution of returns of a policy for a given environment (Section 3.2). This highly expressive distributional representation allows us to both estimate appropriate risk measures and to incorporate them in final decision-making. However, DRL approaches are typically limited to modelling aleatory uncertainty, with epistemic uncertainty due to partial information not being explicitly modelled in terms of the return distribution. We us a bootstrapping [Efron and Tibshirani, 1985] framework to represent epistemic uncertainty. Our framework, which we call SENTINEL-K, is illustrated in Figure 1. At a high level, we use Categorical Deep Q Network (CDQN) [Bellemare et al., 2017] to model aleatory uncertainty and a bootstrapped ensemble for epistemic uncertainty. These can be used with any coherent measures and ensemble algorithm.

We discuss related work in Section 2. This is followed by some background on risk measures, Markov decision processes, and DRL in Section 3. SENTINEL-K is flexible enough to use any combination of coherent risk measures for aleatory and epistemic risks, as we explain in Section 4. The algorithm is described in detail in Section 5, with Section 5.1 and 5.2 showing how the ensemble is created and its members weighted respectively.

Section 6 examines the performance of SENTINEL-K with a composite CVaR metric on a highway environment with 10 cars. Our results show that our approach leads to fewer number of crashes than competing algorithms: Variational DQN (VDQN) [Tang and Agrawal, 2018], CDQN [Bellemare et al., 2017], total variance decomposition Uncertainty

Aware-DQN (UA-DQN) [Clements et al., 2019], as well as SENTINEL-K with additive CVaR estimate, which we used as an ablation test to showcase the importance of the using a coherent composite risk. The supplementary material includes further experiments, showing that SENTINEL-K features significantly improved estimates of return distributions, and shows that using FTRL for weighing the ensemble members measurably improves performance.

## 2 RELATED WORK

For RL applications in the real world, such as for autonomous driving and robotics, *risk-sensitive* RL approaches can avoid the negative consequences of excessive exploration that may lead to unsafe decisions in real-life. This has initiated a spate of research efforts [Howard and Matheson, 1972, Satia, 1973, Coraluppi and Marcus, 1999, Marcus et al., 1997, Mihatsch and Neuneier, 2002, Prashanth and Fu, 2018] spanning five decades. But the majority of risk-sensitive RL papers [Howard and Matheson, 1972, Coraluppi and Marcus, 1999, Marcus et al., 1997] focused on discrete state-space MDPs and either aleatory or epistemic risk. We are interested in designing a general risk-sensitive framework applicable to any type of state space and risk.

Both *aleatory* and *epistemic* uncertainties are important for risk-sensitive RL. The former expresses the *randomness* inherent to the problem and the latter a *lack of knowledge* about the problem. Aleatory risk-sensitivity in MDPs was first considered by [Howard and Matheson, 1972], who in-

troduced the idea of exponential utilities for the return.[1] Epistemic uncertainty in MDPs was investigated by [Satia, 1973], who provided game theoretic and Bayesian solution methods. Later works [Coraluppi and Marcus, 1999, Marcus et al., 1997, Mihatsch and Neuneier, 2002] extend risk-neutral methods to the risk-sensitive setting by using a non-linear utility [García and Fernández, 2015]. They consider aleatory risk-sensitive RL with exponential utility on the return [Mihatsch and Neuneier, 2002]. Follow-up works [Chow and Ghavamzadeh, 2014, C. et al., 2015] focus on scaling up these approaches. Other work on risk-sensitive RL focuses on CVaR [Chow and Ghavamzadeh, 2014, Tamar et al., 2015, Chow et al., 2015]. There have been recent works considering epistemic risk [Eriksson and Dimitrakakis, 2020], wherein problem uncertainty is expressed in a Bayesian framework as a distribution over MDPs. Depeweg et al. [2018], Clements et al. [2019] intuitively incorporates both of these risks in decision making. Depeweg et al. [2018] considers the risk in the per-step rewards obtained in a MDP while Clements et al. [2019] proposes to use the additive formulation of epistemic and aleatory risks. Both of them use variance, which is not a coherent measure [Artzner et al., 1999]. Unlike previous work, our methodology of composite risk also allows us to apply any pair of coherent risk measures[2] to aleatory and epistemic uncertainty.

We instead define a generalised composite risk measure that takes into account both epistemic and aleatory uncertainty, and their entangled effect. Coherence is important, as we show that for any two coherent risk measures the composite risk retains coherence. This gives a principled approach for combining different application-appropriate risk measures for epistemic and aleatory uncertainties.

To express aleatory uncertainty, we rely on a distributional RL method called CDQN, which incorporates highly expressive approximators to model continuous and multimodal return distributions. In addition, we leverage ensemble methods to express epistemic uncertainty. Ensemble methods have first been used in risk-neutral RL by for representing epistemic uncertainty in order to improve exploration [Dimitrakakis, 2006, 2007]. This approach was later applied to MDPs by Osband et al. [2016]. On the other hand, Wiering and Van Hasselt [2008] used ensembles to combine policies instead. Ensembles have also been used to represent aleatory [Faußer and Schwenker, 2015, Pacchiano et al., 2020] uncertainty. Recently, [Depeweg et al., 2018, Clements et al., 2019] also use multiple Bayesian Neural Networks (BNNs) to estimate epistemic uncertainty. In the best of our knowledge, we are the first to use bootstrapped CDQNs for quantifying epistemic risk, which gives us freedom to model distributions on plausible MDPs without any

---

[1]Here, we use return to mean the total discounted reward

[2]For example, CVaR, Wang risk measure [Wang, 2002], Standard Deviation (SD).

structural assumptions, e.g. Gaussian distribution on parameters of Bayesian NNs or Gaussian distribution on state transitions [Clements et al., 2019]. An additional difference with prior work is that we use a follow the regularised leader (FTRL) algorithm to weigh the ensemble members in order to improve our uncertainty estimates.

## 3  BACKGROUND

### 3.1  RISK MEASURES: COHERENCE

The idea of quantifying risk in decision making is long-studied in decision theory and has found multiple applications in finance and actuarial science. A *risk measure* maps a real-valued distribution to a real number, and quantifies the probability of occurrence of an event away from the expectation [Szegö, 2002]. Some well-known risk measures are variance, Value at Risk (VaR) and Conditional Value at Risk (CVaR). *Coherent* risk measures obey a set of axioms Artzner et al. [1999]: normalisation, monotonicity, subadditivity, homogeneity, and translation invariance. Not all risk measures are coherent: CVaR is coherent, but variance and VaR do not satisfy respect homogeneity and subadditivity respectively [Artzner et al., 1999].

If a coherent risk measure also satisfies comonotonic subadditivity [Song and Yan, 2009, Axiom 4], it can be expressed as an expectation over a distorted distribution function, for a concave *distortion function* $U_\alpha : [0, 1] \rightarrow [0, 1]$. Specifically (see [Wang et al., 1997, Theorem 2]) a random variable $Z$ with associated probability measure $P$ and cumulative distribution function $F_Z$ satisfies:

$$\text{Risk}_{U_\alpha}(Z) \triangleq \int_{\mathcal{Z}} Z \, \mathrm{d}(U_\alpha \circ P)$$
$$= \int_{\mathcal{Z}} U_\alpha(1 - F_Z(z)) \, \mathrm{d}z = \int_0^1 U_\alpha(t) \, \mathrm{d}q(1 - t), \quad (1)$$

where $(U_\alpha \circ P)(A) \triangleq U_\alpha[P(A)]$ for any $A \subseteq \mathcal{Z}$. The last line is obtained from substitution of variables [Wirch and Hardy, 2001]. Here, $q$ is the quantile function, i.e. $q(1-t) = \inf\{z \geq 0 | F_Z(z) \geq 1 - t\} = F_Z^{-1}(1 - t)$, $U(0) = 0$, and $U(1) = 1$. Since in this paper we use the risk measures for decision making, we represent a coherent risk measure through its corresponding *distortion function* $U_\alpha$.

In this paper we focus on the *CVaR* [Rockafellar et al., 2000] risk measure. It is extensively used in risk-sensitive RL as it is coherent, applies to general $L_p$ spaces, and captures the heaviness of the tail of a distribution. It is the expectation of the worst $\alpha$-quantile of a probability distribution, with $\alpha \in [0, 1]$:

$$CVaR_\alpha(Z) \triangleq \mathbb{E}[Z \mid Z \leq \nu_\alpha \wedge \mathbb{P}(Z \geq \nu_\alpha) = 1 - \alpha]. \quad (2)$$

For CVaR, $U_\alpha(t) = \min\{\frac{t}{1-\alpha}, 1\}$, For $\alpha = 1$, CVaR reduces to the expected value, and thus risk-neutrality.

Due to generality of our methodology and the composite risk formulation, we are able to incorporate other coherent risk measures such as the Wang risk measure [Wang, 2002], and standard deviation [Cirillo, 2017] (Fig. 4).

## 3.2 RL: MDP AND DISTRIBUTIONAL RL

**MDPs.** We consider problems that can be modelled by a Markov Decision Process (MDP) [Sutton and Barto, 2018]. An MDP is a tuple $\mu \triangleq (\mathcal{S}, \mathcal{A}, \mathcal{R}, \mathcal{T}, \gamma)$. $\mathcal{S} \in \mathbb{R}^d$ is a state space of dimension $d$. $\mathcal{A}$ is the set of admissible actions. $\mathcal{T}$ is a transition kernel that determines the probability of successor states $s'$ given the present state $s$ and action $a$. The reward function $\mathcal{R}$ quantifies the goodness of taking action $a$ in state $s$. In the risk-neutral setup, the goal of the agent is to find a policy $\pi : \mathcal{S} \to \mathcal{A}$ to maximise expected value of cumulative rewards given a time horizon $T$: $V^\pi(s, a) = \mathbb{E}\left[\sum_{t=0}^{T} \gamma^t R(s_t, a_t)\right]$. Here, $s_t \sim \mathcal{T}(.|s_{t-1}, a_{t-1})$, $a_t = \pi(s_t)$, $s_0 = s$, $a_0 = a$, and the discount factor $\gamma \in (0, 1)$.
**Distributional RL.** The variable at the core of both risk-neutral and risk-sensitive RL is usually the accumulated discounted reward $Z^\pi(s, a) \triangleq \sum_{t=0}^{T} \gamma^t R(s_t, a_t)$. $Z^\pi(s, a)$ is called the return of a policy $\pi$. In distributional RL, the goal is to learn the return distribution $Z^\pi(s, a)$ obtained by following policy $\pi$ from state $x$ and action $a$ under the given MDP.

In this work, we choose to extend CDQN by Bellemare et al. [2017], as it permits richer representations of distributions, and flexibility to compute different statistics. The intuition of using this distributional framework for risk-sensitive RL is its flexibility to model multimodal and asymmetrical distributions, which is important for an accurate estimate of risk.

# 4 QUANTIFYING COMPOSITE RISK

In risk-sensitive RL, we encounter two types of uncertainties: *aleatory* and *epistemic*. Aleatory uncertainty is engendered by the stochasticity of the MDP model $\mu$ and the policy $\pi$. Epistemic uncertainty exists due to the fact that the MDP model $\mu$ is unknown. In the Bayesian setting, this is represented as a belief distribution $\beta$ over a set of plausible MDPs $\Theta$. Hence, risk measures can also be defined with respect to the MDP distribution. Consequently, as an agent learns more about the underlying MDP, the epistemic risk vanishes. The aleatory risk is inherent to the MDP $\mu$ and policy $\pi$, and thus persists even after correctly estimating the model $\mu$. Let us now define risk measures for aleatory and epistemic uncertainties, and then combine them into a composite risk measure.

**Aleatory Risk.** Given a coherent risk measure with distortion function $U_\alpha^A$, the aleatory risk is quantified as the deviation of total risk of individual models from the risk of the average model.

$$A(U_\alpha^A, \beta) \triangleq \int_\Theta \int_\mathcal{Z} Z \, \mathrm{d}(U_\alpha^A \circ \mathbb{P})(Z|\theta) \, \mathrm{d}\beta(\theta)$$
$$- \int_\Theta \int_\mathcal{Z} \hat{Z} \, \mathrm{d}(U_\alpha^A \circ \mathbb{P})(\hat{Z})$$

Here, $\mathbb{P}(\hat{Z}) = \int_\Theta \mathbb{P}(Z|\theta) \, \mathrm{d}\beta(\theta)$, i.e. the return distribution of the average model. The centered definition of aleatory risk is necessary to show that additive risk is a special case of composite risk.

**Epistemic Risk.** Given a coherent risk measure with distortion function $U_\alpha^E$, the epistemic risk quantifies the uncertainty invoked by not knowing the true model. Thus, the risk can be computed over any statistics of the models, such as expectation.

$$E(U_\alpha^E, \beta) \triangleq \int_\Theta \int_\mathcal{Z} Z \, \mathrm{d}\mathbb{P}(Z|\theta) \, \mathrm{d}(U_\alpha^E \circ \beta)(\theta)$$

**Composite Risk under Model and Inherent Uncertainty.** In typical risk-sensitive RL settings, the true MDP model is both unknown and inherently stochastic. Thus, the overall uncertainty is a composition of aleatory and epistemic uncertainties. For that reason, quantify it using what we call the *composite risk*.

**Definition 1** (Composite Risk). *For two coherent risk measures with distortion functions $U_{\alpha_1}^A$ and $U_{\alpha_2}^E$, belief distribution $\beta$ on model parameters $\theta \in \Theta$, and a random variable $Z \in \mathcal{Z}$, the composite risk of epistemic and aleatory uncertainties is defined as*

$$F^C(U_{\alpha_1}^A, U_{\alpha_2}^E, \beta) \triangleq \mathrm{Risk}_{U_{\alpha_2}^E}\left(\mathrm{Risk}_{U_{\alpha_1}^A}(Z|\theta)|\beta\right)$$
$$= \int_\Theta \int_\mathcal{Z} Z \, \mathrm{d}(U_{\alpha_1}^A \circ \mathbb{P})(Z|\theta) \, \mathrm{d}(U_{\alpha_2}^E \circ \beta)(\theta)$$
$$= \int_0^1 \int_0^1 U_{\alpha_2}^E(v) U_{\alpha_1}^A(u) \, \mathrm{d}q_{Z|\theta}(1-u) \, \mathrm{d}q_\beta(1-v) \quad (3)$$

Here, $q_{Z|\theta}$ and $q_\beta$ are quantile functions of $Z$ conditioned on $\theta$ and that of $\theta$ respectively. For brevity, we also denote $F^C(U_{\alpha_1}^A, U_{\alpha_2}^E, \beta)$ as $\mathrm{Risk}_{U_{\alpha_2}^E} \circ \mathrm{Risk}_{U_{\alpha_1}^A}$ (e.g. CVaR $\circ$ CVaR), whenever it is clear from the context.

**Theorem 2** (Coherence). *If $U_{\alpha_1}^A$ and $U_{\alpha_2}^E$ are distortion functions for two coherent risk measures, the composite risk measure $F^C(U_{\alpha_1}^A, U_{\alpha_2}^E, \beta)$ is also coherent.*

The proof of Theorem 2 is available in Supplementary material. The generic nature of our composite risk definition allows us to use different risk measures compatible with epistemic and aleatory risks. This is demonstrated in experiments (Figure 4) using different combinations of CVaR, Wang risk, and standard deviation for quantifying epistemic and aleatory uncertainties. This flexibility was absent in previous risk-sensitive RL literature [Eriksson and Dimitrakakis, 2020, Depeweg et al., 2018, Clements et al., 2019].

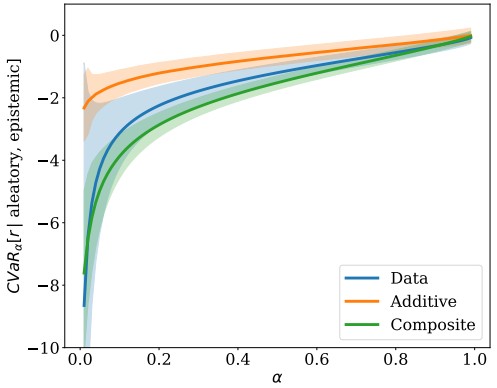

Figure 2: Estimation of total $CVaR_\alpha$ from a mixture of 100 Gaussians sampled from a posterior distribution. Total $CVaR_\alpha[Data]$ is based on the marginal distribution of $r$ as in Example 1. We compare this with composite and additive estimates and illustrate results over 100 runs. Here, lower value of CVaR indicates higher mass on the left tail of the distribution and higher risk of obtaining low returns.

**Comparison with Additive Risk Formulations.** Clements et al. [2019], Depeweg et al. [2018] use a weighted sum of epistemic and aleatory variances as their risk measure. This formulation has mainly two problems. First, variance is not a coherent risk measure as it does not follow the homogeneity and subadditivity properties, as shown in [Cirillo, 2017]. Secondly, we show that even if we replace the variance with a coherent risk measure, the additive formulation is equivalent to considering $U_\alpha^E$ as an identity function. Thus, it is less sensitive to the effect of epistemic uncertainty than composite risk. More formally:

**Theorem 3.** *We are given two sources of aleatory and epistemic uncertainties $\xi_1$ and $\xi_2$. If $U_{\alpha_1}^A$ and $U_{\alpha_2}^E$ are distortion measures for two coherent risk measures quantifying aleatory and epistemic risks respectively, then, i) $F^A(U_{\alpha_1}^A, \beta) = F^C(U_{\alpha_1}^A, I, \beta)$, where $I$ is the identity function, and ii) $F^C(U_{\alpha_1}^A, U_{\alpha_2}^E, \beta) \geq F^A(U_{\alpha_1}^A, \beta)$, if $\alpha_2 \neq 1$.*

**Example 1** (A Reductive Empirical Evaluation of Composite and Additive Risks)**.** *We consider a mixture of 100 Gaussians: $p(r) = \sum_{i=1}^{100} \phi_i \mathcal{N}(\mu_i, \sigma_i^2)$, where $\Phi \sim Dir([0.5]^{100}), \mu \sim \mathcal{N}(0, 1)$, and $\sigma^2 \sim \Gamma^{-1}(2, 0, 1)$. We compute $CVaR_\alpha[r]$ using the data generated from this mixture over 100 runs. We further estimate composite risk with $U_E, U_A = CVaR_\alpha$ and additive risk with $U_A = CVaR_\alpha$. The results illustrated in Figure 2 show that the additive CVaR risk strictly underestimates the total CVaR risk computed from the data, whereas the composite risk is closer to the one computed from data. Specifically, for lower values of $\alpha$ (specifically, $\alpha \leq 0.5$), i.e. towards the extreme end of the left tail where events occur with low probability, the additive CVaR risk deviates significantly from data whereas the*

*composite measure yields closer estimation. Such values of $\alpha$'s are typically interesting for risk-sensitive applications.*

This means that for given sources of aleatory and epistemic uncertainties the additive risk which only considers expectation over epistemic uncertainty will always underestimate the composite effect of epistemic risk. Thus, we observe that additive risk leads to worse risk-sensitive performance than composite risk in RL problems (Table 1 and Figure 3).

# 5 ALGORITHM: SENTINEL-K

Now, we outline the algorithmic details of SENTINEL-K that estimates composite risk over returns using an ensemble of $K$ distributional RL estimators, namely CDQN, in tandem with an adaptation of FTRL for estimator selection, and leverage the estimates for decision making.

**Sketch of the Algorithm.** Pseudocode of SENTINEL-K with composite risk is given in Algorithm 1. It has two main blocks: obtaining $K$ estimates of return distribution with distributional RL framework (Lines 4- 13), and using them to compute composite risk for each action (Lines 15- 21). Finally, following the mechanism of Q-learning [Watkins and Dayan, 1992], it chooses the action with maximal composite risk in the decision making step (Line 23).

In the first block (Lines 4- 13), we specifically use an ensemble of $K$ CDQNs. Each CDQN uses target and value networks for estimating the return distribution. We set a schedule for updating the target networks $\Gamma_1$ and a more frequent one ($\Gamma_1 \cup \Gamma_2$) for the value networks (Section 5.1).

The second block (Lines 15- 21) is used for decision-making and iterated at every time step. It adapts the FTRL algorithm (Section 5.2) for aggregating the $K$ estimated return distributions and to compose aleatory risk $Q_i^A(s_t, a)$ of each of the estimators to provide a final estimate of the composite risk $Q^C(s_t, a)$ for each action, and then selecting the action with highest $Q^C(s_t, a)$.

## 5.1 ENSEMBLING AND BOOTSTRAPPING $K$-ESTIMATORS

The ensemble of SENTINEL-K consists of $K$ distribution estimators. Each estimator gets its own dataset $\{D_i\}_{i=1}^K \subseteq \mathcal{D}$, value network $\{\theta_i\}_{i=1}^K$ and target network $\{\theta_i^-\}_{i=1}^K$. The $K$ datasets are created from the original dataset $\mathcal{D}$ by *data masking* (Line 5). For each transition $s_t, a_t, r_t, s_{t+1}$, a fixed weight vector $\mathbf{u}_t \in [0, 1]^K$ is generated such that $u_t^j \sim Ber(\frac{1}{3})$. Thus, on an average, each estimator $i$ has access to $\frac{1}{3}$ of the dataset. Details about data masking are in Supplementary material.

After preparing the datasets for the estimators, the target and value networks of the CDQN have to be updated and

optimised. For $i$-th estimator, it begins with sampling mini batches of data $\tau$ from the respective dataset $D_i$ (Line 7). Then, this dataset is used to compute the composite risk for all actions $a \in \mathcal{A}$ and to obtain $a^*$ (Lines 8- 9). Obtaining the composite risk first involves estimating the aleatory risk with $Q_i^A(s_t, a) = \int_{\mathcal{Z}} Z \, \mathrm{d}(U_{\alpha_1}^A \circ \mathbb{P})(Z|\theta_i)$ for a particular estimator $i$. This quantity can be attained by considering each of the estimators separately, however, as we turn to compute the epistemic risk the estimators jointly contribute to this risk. Then, we compose the aleatory risk of all the estimators to compute $Q^C(s_t, a) = \mathrm{Risk}_{U_{\alpha_2}^E}(\{Q_i^A(s_t, a)\}_{i=1}^K)$. Here, $\mathrm{Risk}_{U_{\alpha_2}^E}$ is the risk measure corresponding to the distortion $U_{\alpha_2}^E$. Finally, the optimal action $a^* = \arg\max_a Q^C(s_t, a)$, and the risk estimates $Q^C(s_t, a)$ are used to update the value and network parameters $\{\theta_i\}_{i=1}^K$ and $\{\theta_i^-\}_{i=1}^K$ (Lines 10-11) by minimising the cross-entropy loss of the current parameters and the projected Bellman update as described in [Bellemare et al., 2017].

Ensembling estimators have been shown to outperform individual estimators as seen in [Wiering and Van Hasselt, 2008, Faußer and Schwenker, 2015, Osband et al., 2016, Pacchiano et al., 2020]. Further, incorporating multiple estimators introduces uncertainty over the estimators. Because of having separate data sets, each of the estimators learn different parts of the MDP. Thus, uncertainty over estimators acts as a quantifier of the model uncertainty. In Section 6, we show that this ensemble-based approach leads SENTINEL-K to achieving superior performance.

## 5.2 WEIGHING ESTIMATES WITH FTRL

Now, the question is to adaptively and accurately aggregate the $K$ estimated return distributions. Pacchiano et al. [2020] shows that adaptive model selection can boost performance in comparison to model averaging. The rationale for this can be given by seeing that some estimators might be overly optimistic or pessimistic. By weighing these less, you can effectively have a more robust ensemble. Further discussion of this issue is given in Supplementary material.

We adapt the Follow The Regularised Leader (FTRL) algorithm [Cesa-Bianchi and Lugosi, 2006] studied in bandits and online learning for adaptively weighing the estimators. FTRL puts exponentially more weight on an estimator depending on its accuracy of estimating the return distribution. Since we do not know the 'true' return distribution, we use the KL-divergence from the posterior of a single estimator $i$, $\mathbb{P}(Z|\theta_i)$, to the posterior marginalised over $\beta(\theta)$, i.e. $l(\theta_i, \beta) \triangleq D_{\mathrm{KL}}\left(\mathbb{P}(\hat{Z}) \,\|\, \mathbb{P}(Z|\theta_i)\right)$, as proxy of estimation loss of estimator $i$. FTRL selects estimator $i$ with weight

$$w_i = \frac{e^{\lambda l(\theta_i, \beta)}}{\sum_j e^{\lambda l(\theta_j, \beta)}}, \quad \lambda \in [0, \infty). \qquad (4)$$

Using FTRL weights for aggregating the $K$ return distri-

butions is analogous to using an exponentially weighted average forecaster [Cesa-Bianchi and Lugosi, 2006] on the $K$ learners to create a final estimate of the return distribution and corresponding composite risk. This leads to a better aggregation of individual estimates than equally weighted average or a greedy selection of the best estimate [Cesa-Bianchi and Lugosi, 2006, Theorem 2.2]. Having computed the weights $\mathbf{w}$ (Line 16), we compute the weighted composite risk measure by first computing the aleatory risk of each of the estimators, $Q_i^A(s_t, a) = \int_{\mathcal{Z}} Z \, \mathrm{d}(U_{\alpha_1}^A \circ \mathbb{P})(Z|\theta_i)$ (Line 18), and then the composite risk is computed by $Q^C(s_t, a) = \mathrm{Risk}_{U_{\alpha_2}^E}(\{w_i Q_i^A(s_t, a)\}_{i=1}^K)$ (Line 20). Here, $\lambda \in [0, \infty)$ is a regularising parameter that determines to what extent estimators far away from the marginal estimator should be penalised. If $\lambda \to 0$, we obtain standard model averaging. If $\lambda \to \infty$, it reduces to greedy selection. We experimentally show that performing FTRL with a reasonable $\lambda$ value, namely 1, leads to better performance.

**Action Selection.** The algorithm always selects the action with the high composite risk $Q^C$. Its behaviour depends on the choice of risk measures or distortion utility functions $U_{\alpha_1}^A$ and $U_{\alpha_2}^E$. SENTINEL-K reduces to a risk-neutral algorithm if we choose both $U_{\alpha_1}^A, U_{\alpha_2}^E$ as identity functions, and to additive risk-sensitive algorithm if we choose $U_{\alpha_2}^E$ as identity. Designing it to accommodate composite risk provides us the flexibility to be risk-sensitive, risk-neutral, and treating epistemic and aleatory risk with different metrics.

## 6 EXPERIMENTAL EVALUATION

We test the risk-sensitive performance of SENTINEL-K with composite CVaR risk in two environments with continuous state spaces. We also display the flexibility of our composite risk formulation by evaluating heterogeneous risks with SENTINEL-K.[3] Settings for each of these experiments and results are elaborated in corresponding subsections. In all the experiments, we use 4 CDQNs in the ensemble and call it SENTINEL-4. We justify this choice of $K = 4$ in Supplementary material. For each experiment, we report the mean and standard error of the mean over 20 runs for $10^5$ steps.

**Risk-sensitive Performance.** In order to demonstrate performance in a larger domain, we opt to evaluate SENTINEL-4 in the *highway* [Leurent, 2018] environment. Highway is an environment developed to test RL for autonomous driving. We use a version of the *highway-v1* domain with five lanes, and ten vehicles in addition to the ego vehicle. In this environment, the episode is terminated if any of the vehicles crash or if the time elapsed is greater than 40 time steps. The reward function is a combination of multiple factors, including staying in the right lane, the ego vehicle speed,

---

[3] Ablation studies for risk-neutral SENTINEL are in Appendix.

Table 1: Performance of risk-neutral (VDQN, CDQN, SENTINEL-K), aleatory risk-sensitive VDQN-CVaR, UA-DQN and risk-sensitive (SENTINEL-4 with additive and composite CVaRs) for highway-v1 with 10 vehicles. Results are reported over 20 runs. SENTINEL-4 with composite CVaR performs better.

| Agent | Value $\pm\sigma$ | Aleatory metric $\pm\sigma$ | # crashes $\pm\sigma$ |
|---|---|---|---|
| $\text{VDQN}_{RN}$ Tang and Agrawal [2018] | $23.30 \pm 0.36$ | $14.29 \pm 0.80$ | $1252.33 \pm 170.35$ |
| $\text{CDQN}_{RN}$ Bellemare et al. [2017] | $25.96 \pm 0.51$ | $19.50 \pm 1.44$ | $839.53 \pm 150.20$ |
| $\text{SENTINEL-4}_{RN}$ | $26.56 \pm 0.32$ | $20.88 \pm 1.25$ | $617.11 \pm 100.15$ |
| $\text{VDQN-CVaR}_A$ Tang and Agrawal [2018] | $24.39 \pm 0.50$ | $16.64 \pm 1.25$ | $871.33 \pm 171.23$ |
| $\text{UA-DQN}_{E+A}$ Clements et al. [2019] | $24.46 \pm 0.29$ | $16.9 \pm 0.44$ | $1060.65 \pm 13.94$ |
| $\text{SENTINEL-4}_{E+A}$ | $26.82 \pm 0.42$ | $21.54 \pm 1.40$ | $645.55 \pm 127.59$ |
| $\text{SENTINEL-4}_{E\circ A}$ | $\mathbf{27.43 \pm 0.13}$ | $\mathbf{24.16 \pm 0.54}$ | $\mathbf{341.18 \pm 43.86}$ |

---

**Algorithm 1** SENTINEL-K with Composite Risk

1: **Input:** Initial state $s_0$, action set $\mathcal{A}$, distortion measures $U^A_{\alpha_1}, U^E_{\alpha_2}$, hyperparameter $\lambda$, target networks $[\theta^-_1, ..., \theta^-_K]$, value networks $[\theta_1, ..., \theta_K]$, update schedule $\Gamma_1, \Gamma_2$.
2: **for** $t = 1, 2, \ldots$ **do**
3:   //* Update $K$-value and target networks for estimating return distributions *//
4:   **for** $t' \in \Gamma_1 \cup \Gamma_2$ **do**
5:     Generate $\{D_1, ..., D_K\} \leftarrow \text{DataMask}(\mathcal{D}^{t'})$
6:     **for** $i = 1, \ldots, K$ **do**
7:       Sample mini batch $\tau \sim D_i$
8:       Estimate (3) $F^C(Z(s_t, a)|U^A_{\alpha_1}, U^E_{\alpha_2}, \beta)$ using $\tau$ and $K$-target networks $\{\theta^-_i\}^K_{i=1}$.
9:       Get $a^* = \arg\max_a F^C(Z(s_t, a)|U^A_{\alpha_1}, U^E_{\alpha_2}, \beta)$
10:       Update value network $\theta_i$ using $\tau, a^*$
11:       Update target network $\theta^-_i$ using $\tau, a^*$ if $t' \in \Gamma_1$
12:     **end for**
13:   **end for**
14:   //* Estimate the composite risk of each action using the estimated return distributions *//
15:   **for** $a \in \mathcal{A}$ **do**
16:     Compute weights $\mathbf{w} = w_1, ..., w_K$ from Eq. 4.
17:     **for** $i$ **in** $K$ **do**
18:       Compute aleatory risks $Q^A_i(s_t, a)$ from $\int_{\mathcal{Z}} Z \, d(U^A_{\alpha_1} \circ \mathbb{P})(Z|\theta_i)$
19:     **end for**
20:     Compute composite risk over weighted aleatory estimates $Q^C(s_t, a) = \text{Risk}_{U^E_{\alpha_2}}(\{w_i Q^A_i(s_t, a)\}^K_{i=1})$
21:   **end for**
22:   //* Action selection *//
23:   Take action $a_t = \arg\max_a Q^C(s_t, a)$
24:   Observe $s_t$ and update the dataset $\mathcal{D}^t \leftarrow \mathcal{D}^{t-1} \cup \{s_t, a_{t-1}, s_{t-1}, r_{t-1}\}$
25: **end for**

---

and the speed of the other vehicles.

We test the risk-neutral CDQN and VDQN algorithms, an aleatory risk-sensitive VDQN and the total variance decomposition algorithm UA-DQN along with SENTINEL-4 with both additive and composite CVaRs. The typical performance metric for this scenario is the expected discounted return $\mathbb{E}^\pi_\mu[R]$. In order to test the risk-sensitive performance, we use two metrics. In order to measure aleatory risk $U^A_{\alpha_1}[R \mid \pi, \mu]$, we use CVaR as $U^A_{\alpha_1}$ with threshold $\alpha = 0.25$. The CVaR metric is a statistic of the left-tail of the return distribution and higher values would mean better performance in the $25\%$ worst-cases of performance. Finally, as a proxy for the epistemic risk, we use the number of crashes (lower is better).

Experimental results are illustrated in Table 1 and Figure 3. From Table 1, we observe that our algorithm with composite risk achieves a higher value, higher estimate of aleatory risk, and less number of crashes. Thus, SENTINEL-4 with composite CVaR outperforms the competing algorithms in terms of all three metrics. The simultaneous improvement in both the value function and #crashes is due to the fact that *highway* is designed to have a reward function that penalises unsafe driving. Additionally, we observe that the variance of performance metrics over 20 runs is the least for our algorithm with composite CVaR measure. This shows the stability of our algorithm which is another demonstration of good risk-sensitive performance. Figure 3 resonates with these observations in terms of the total number of crashes.

**Heterogeneous Risk Measures.** In order to demonstrate the flexibility of the composite risk framework estimated with SENTINEL, we investigate performance using heterogeneous coherent risk measures, that composes different coherent risk measures for aleatory and epistemic risk. The chosen risk measures are aleatory and epistemic CVaR, aleatory and epistemic Wang risk, aleatory CVaR with epistemic standard deviation, and aleatory standard deviation with epistemic CVaR. Note that any combination of coherent risk measures is possible. We evaluate SENTINEL-4 in the *CartPole-v0* environment [Brockman et al., 2016]. This environment is a popular test-bed for continuous state-space RL tasks. In the environment, a reward of 1 is attained for every time step the pole is kept upright. If the pole falls to either of the sides or if the number of time steps reaches 200, the episode is terminated. This means that the undiscounted return attained per episode is in $[0, 200]$. Thus, we

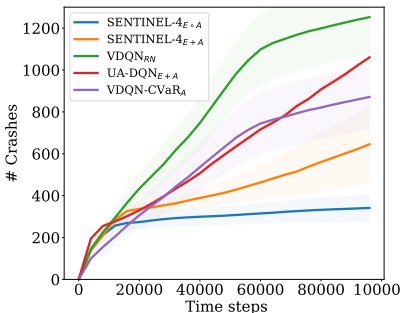
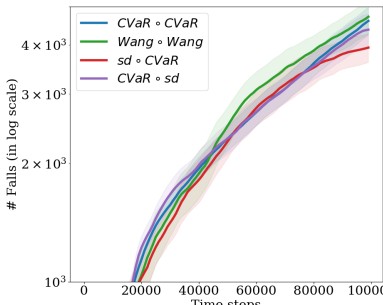
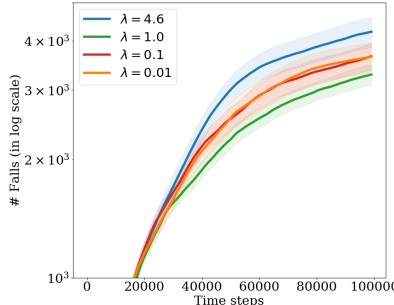

Figure 3: The total number of crashes in highway environment with $10$ vehicles over $20$ runs and horizon $10^6$. Fewer #crashes indicate better risk-sensitive performance.

Figure 4: Performance and convergence of SENTINEL-4 using different risk measures. We show the number of falls in the *CartPole* environment over $20$ runs with different initialisation.

Figure 5: Performance and convergence of SENTINEL-4 (risk-neutral) for different values of $\lambda$. We show the number of falls in *CartPole* environment over $20$ runs with different initialisation.

choose $V_{min} = 0, V_{max} = \frac{1-\gamma^{200}}{1-\gamma}$ as the histogram support of CDQN. The results are shown in Figure 4, which demonstrates than SENTINEL-4 performs flexibly and comparably for these composite risks.

**FTRL vs. Average vs. Greedy.** We choose $[0.01, 0.1, 1.0, \ln 100]$ as the different values of the regularising hyperparameter $\lambda$ and test the performance of SENTINEL-4 for *CartPole-v0*. As $\lambda \to 0$, we perform standard model averaging which is sensitive to outliers. As $\lambda \to \infty$, model selection gets greedily biased towards the best average estimator while not providing other estimators a chance to improve. A sound value of $\lambda$ would be one that excludes outlier estimators while still involves most of the other estimators. Figure 5 shows performance in terms of cumulative # Falls (lower is better) for the $\lambda$ values with $CVaR_{0.25} \circ CVaR_{0.25}$. We observe that FTRL with reasonable $\lambda = 1.0$ shows better performance, i.e. less number of falls, than the ones with large $\lambda = 4.6$ and small $\lambda$'s $0.01$ and $0.1$. We also observe that for $\lambda = 1$ the variance of #Falls is significantly less than that of other values and thus, stability of performance.

**Summary of Results.** Fig. 3 shows the risk-sensitive performance of VDQN, CDQN, aleatory CVaR, total variance decomposition UA-DQN and SENTINEL-4 additive and composite CVaR risks on a large continuous state environment. SENTINEL-4 with composite risk outperforms competing algorithms in terms of the achieved value function and estimated aleatory risk. It causes the least number of crashes than competing algorithms. Fig. 4 demonstrates the ability to chose any coherent risk measure for SENTINEL-K, including different risk measures for both epistemic and aleatory risk. Fig. 5 shows that selecting $\lambda$ is important in bootstrapped RL, and tuning it yields better performance over model averaging ($\lambda \to 0$) and greedy selection ($\lambda \to \infty$). We defer the results on the choice of $K$ in ensemble, convergence in return distribution, and improved efficiency in estimating multi-modal return distributions, to Appendix.

## 7 DISCUSSION

In this paper, we study the problem of risk-sensitive RL. We propose two main contributions. The first is the *composite risk* formulation that quantifies the holistic effect of aleatory and epistemic risk involved in learning. With a reductive experiment, we show that composite risk estimates the total risk involved in a problem more accurately than existing additive formulations. The second one is *SENTINEL-K* which ensembles $K$ distributional RL estimators, namely CDQNs, to provide an accurate estimate of the return distribution. We adopt FTRL from bandit literature as a means of model selection. FTRL weighs each estimator adaptively and leads to better experimental performance than greedy selection and model averaging. Experiments show that SENTINEL-K achieves superior risk-sensitive performance while used with composite CVaR estimate, and can operate on composition of different risks unlike existing works.

Motivated by the experimental success, we aim to investigate theoretical properties of FTRL-driven bootstrapped distributional RL with and without composite risk estimates.

**Acknowledgements**

We would like to thank Dapeng Liu for fruitful discussions in the beginning of the project, further, this work was partially supported by the Wallenberg AI, Autonomous Systems and Software Program (WASP) funded by the Knut and Alice Wallenberg Foundation and the computations were enabled by resources provided by the Swedish National Infrastructure for Computing (SNIC) at C3SE partially funded by the Swedish Research Council through grant agreement no. 2018-05973.

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
