# OpenReview forum: "SENTINEL: Taming Uncertainty with Ensemble based Distributional Reinforcement Learning"
_auai.org/UAI/2022/Conference — UAI 2022 Poster_

### Official Review · Reviewer_jhWV · 2022-04-11

**Q2(1) Originality/Novelty:** 2
**Q2(2) Significance/Impact:** 2
**Q2(3) Correctness/Technical Quality:** 2
**Q2(6) Clarity Of Writing:** 3
**Q6 Overall Score:** 4
**Q8 Confidence In Your Score:** 3

**Q1 Summary And Contributions:**

This paper first proposes a composite risk formulation in order to accurately capture the combined effect of aleatory and epistemic uncertainty for decision-making in RL.  The second proposition concerns SENTINEL-K which ensembles K distributional RL estimators to provide an accurate estimate of the return distribution. Experiments show that SENTINEL_K (K=4) achieves better risk-sensitive performance while used with composite Conditional Value at Risk (CVaR) estimate.

**Q2 Assessment Of The Paper:**

More detailed information regarding each of these aspects is given below:

**Q2(4) Quality Of Experiments (Optional):**

2: Fair: The experimental evaluation is weak: important baselines are missing, or the results do not adequately support the main claims.

**Q2(5) Reproducibility:**

2: Fair: Key resources (e.g., proofs, code, data) are unavailable but key details (e.g., proof sketches, experimental setup) are sufficiently well-described for an expert to confidently reproduce the main results.

**Q3 Main Strengths:**

The proposition of the composite risk is really interesting

**Q4 Main Weakness:**

- The absence of appendices does not allow to verify the proposed technical results
- Some results are just observed and not explained, for instance the value of \delta relative to figure 6, authors mention that λ = 1.0 shows better performance without any intuition or explanation
- The reductive experiment relative to the first part deserves a better development and analysis

**Q5 Detailed Comments To The Authors:**

•	Experiments show that additive risk leads to worse risk-sensitive performance than composite risk in RL problems. How to confirm that it is a particular case of the composite risk when the epistemic risk measure is replaced with expectation as announced in the abstract. This point is not clear.
•	Figure 1 relative to SENTINEL-K is not well explained it should be linked with algorithm 1
•	Example 2 is not an example it corresponds to the “reductive” experiment of the first proposition, this part deservesto be better organized
•	The max value of t in line 2 of algorithm 1 (for t = 1, 2, . . . ) should be defined
•	Add \beta and K as inputs of algorithm 1
•	Define datamask used in algorithm 1
•	Use Figure or Fig. throughout the paper


**Q7 Justification For Your Score:**

- The absence of appendices does not allow to verify the proposed technical results

**Q9 Complying With Reviewing Instructions:**

1: Yes.

---

### Official Review · Reviewer_eAnU · 2022-04-11

**Q2(1) Originality/Novelty:** 3
**Q2(2) Significance/Impact:** 3
**Q2(3) Correctness/Technical Quality:** 3
**Q2(6) Clarity Of Writing:** 4
**Q6 Overall Score:** 8
**Q8 Confidence In Your Score:** 4

**Q1 Summary And Contributions:**

- In this work the authors introduce a new quantification of risk called composite risk. They also present a new algorithm, SENTINEL-K using ensemble bootstrapping and distributional reinforcement learning. Finally they evaluate their method in 2 problem domains and compare against other distributional reinforcement learning methods.


**Q2 Assessment Of The Paper:**

More detailed information regarding each of these aspects is given below:

**Q2(4) Quality Of Experiments (Optional):**

4: Excellent: The experimental evaluation is comprehensive and the results are compelling.

**Q2(5) Reproducibility:**

3: Good: Key resources (e.g., proofs, code, data) are available and key details (e.g., proofs, experimental setup) are sufficiently well-described for competent researchers to confidently reproduce the main results.

**Q3 Main Strengths:**

This work fills a necessary gap in the risk based decision making literature and also highlights the limits of current approaches.

**Q4 Main Weakness:**

Please see major comments below.

**Q5 Detailed Comments To The Authors:**

- Major Comments
Currently, I am more than happy to recommend this paper to be accepted. However, I have a couple of concerns and it would be great if the author could answer these concerns during the rebuttal.

1 ) Utility functions generally are described to define an agents preference for risk. However, in Section 3.1 a distorted utility function is used, where "it can be expressed as an expectation over a distorted distribution function, for a concave distortion function". Usually in risk-sensitive decision making the shape of the utility function can be used to describe the agent's preference for risk. It looks like the agent's preference for risk can be altered by specifying the alpha parameter. What level of error is introduced by having to tune this parameter to determine optimal performance? Is it more efficient than having a utility function specified a priori or is a lot of tuning required to obtain the desired behavior? It would have been interesting to see your algorithm being compared against a CDQN variant that used a risk-aware utility function.

2 ) Example 1 describes a simple experiment to outline the shortcomings of additive CVar. However, no metric is provided to actually compare the distributions in Figure 2. A metric to describe the difference between the additive and composite risk measures from the underlying data would really strengthen this example. While it is obvious the composite method is better, exactly how much better is hard to see.

3 ) Maintaining a number of distributions clearly increases the computational complexity. Therefore, there seems to be a trade-off between the increased performance and the cost of actually running the algorithm. The aglorithm is tested with experiments with pretty simple returns therefore a small number of bins can be used to model the return distributions. It would be interesting to see if the cost of computation increases with more complex returns over wider ranges where a greater number of bins is needed to accurately model the return distributions.

4 ) While the experiments are interesting they are limiting. The experimental domains are not exactly risk aware mdps but rather mdps where the narrative has been changed to motivate applying risk aware algorithms. It would really have been nice if a risk aware mdp like something from finance could have been used to evaluate the algorithm.

5 ) For the Highway environment, according to the source code "The reward is defined to foster driving at high speed, on the rightmost lanes, and to avoid collisions.". It appears the goal of the experiment is to reduce collisions. Is this correct? If not, this section should be motivated differently as it reads as if this is the goal.

Given the reward is a linear combination of each the objectives outlined in the paper, how are collisions considered as a risk-sensitive objective? Do you alter the weighting on that objective to make it more important. If not then it appears that the risk is simply just specified on the scalarised reward which would mean that no specific objective is actually being considered. Then, the risk is considered on a linear combination of all of the rewards. There is nothing wrong with this but it would mean that the results of the other objectives should be reported also or rather just a graph that would include the value similarly to Table 2.

The line "Finally, as a proxy for the epistemic risk, we use the number of crashes (lower is better)." would lead the reader to assume collisions are somehow considered differently than all other objectives in the domain.

The algorithm is just learning a better policy where all objectives are maximised/minimised with respect to the scalarisation and therefore I would expect that the agent also stays in the right lane and drives more optimally when compared to the other algorithms. Overall the algorithm doesn't appear to identify risk with respect to collisions, but rather identifies risk for the linear scalarisation of the combination of all objectives. If this is the case I think it is important this is stated and the results for the other objectives should be presented too, either in the paper or in the appendix.

6 ) The results for the highway experiment (Table 1) when comparing SENTINEL additive and SENTINEL composite are very close when we look at the value. Why is this? Is this also because additive measure underestimate the risk and therefore cannot perform better than composite (as per Figure 2)? I would expect the composite results to be much better, but it looks like using the SENTINEL algorithm results in a performance boost regardless. It really would be great if we could compare the results from the other two objectives also, as I previously mentioned.

7 ) One aspect that I am unsure about is exploration. How do you ensure the algorithm sufficient explores each MDP?

- Minor Comments

- Figure 1 should show how the estimated return distributions and the aleatory risk interact to become composite risk. It is not very clear that they are combined.

- "Fig. 3 shows the risk-sensitive performance of VDQN, CDQN, aleatory CVaR, total variance decomposition UA-DQN and SENTINEL-4 additive and composite CVaR risks on a large continuous state environment. SENTINEL-4". Fig 3 does not show this, rather Table 1 does.

- Why do the results for Fig 4 and Fig 5 begin at 20,000 time steps? If this is because of the log scale then please state this.

- From the highway problem github :: "In this task, the ego-vehicle is driving on a multilane highway populated with other vehicles. The agent's objective is to reach a high speed while avoiding collisions with neighbouring vehicles. Driving on the right side of the road is also rewarded.". However, this work states "The simultaneous improvement in both the value function and #crashes is due to the fact that highway is designed to have a reward function that penalises unsafe driving.". Can you please explain how the highway environment rewards safer driving? Given the objectives are linearly combined I cannot see how this is the case. Perhaps I have missed something obvious as I am not overly familiar with this environment. Or did you alter the reward function?

**Q7 Justification For Your Score:**



- Overall, I found this paper interesting especially because the authors present an novel idea for calculating both epistemic and aleatory risk and highlight issues with current additive methods. Finally, I would like to thank the authors for providing a very interesting paper!

**Q9 Complying With Reviewing Instructions:**

1: Yes.

---

### Official Review · Reviewer_NSgv · 2022-04-16

**Q2(1) Originality/Novelty:** 3
**Q2(2) Significance/Impact:** 3
**Q2(3) Correctness/Technical Quality:** 3
**Q2(6) Clarity Of Writing:** 3
**Q6 Overall Score:** 5
**Q8 Confidence In Your Score:** 3

**Q1 Summary And Contributions:**

This paper combined the measurements of aleatory risk and epistemic risk to form a new composite risk, which is used to consider the prior two uncertainty simultaneously. Besides, this paper proposed a new distributional Reinforcement Learning (RL) method with K isolated models. Also, the final control output was selected by a Follow The Regulazed Leader (FTRL) method. Some empirical studies showed the superiority of the proposed method compared with the previous risk-averse RL methods.

**Q2 Assessment Of The Paper:**

More detailed information regarding each of these aspects is given below:

**Q2(4) Quality Of Experiments (Optional):**

3: Good: The experimental evaluation is adequate, and the results convincingly support the main claims.

**Q2(5) Reproducibility:**

3: Good: Key resources (e.g., proofs, code, data) are available and key details (e.g., proofs, experimental setup) are sufficiently well-described for competent researchers to confidently reproduce the main results.

**Q3 Main Strengths:**

1. Good presentation;
2. Reasonable solution.

**Q4 Main Weakness:**

1. The definition of the measurement was somehow confusing;
2. The solution was complicated and ad-hoc.

**Q5 Detailed Comments To The Authors:**

This paper studied the risk-averse problem that is important in RL scenarios. The combination of the aleatory risk and the epistemic risk was novel. Also, the proposed method is reasonable for solving these two aspects simultaneously. But on the other hand, some concerns also remain:

1. For the definition of the composite risk, it seems like the direct combination of the aleatory risk and the epistemic risk. But where is the second part of the aleatory risk in the composite risk? I think this total risk is not a constant that can be thrown casually. The authors should focus on how the form of composite risk comes.

2. The proposed method is very complicated, especially for the usage of the FTRL part, which makes the previous K target networks isolated from each other. The authors should consider using a more end-to-end method to replace the FTRL part, since you do not really care about the online gradient here. In addition, the usage of the negative entropy as the regularizer is too ad-hoc, since it requires the loss function should be linear, meanwhile the decision domain of $w$ should be simplex, so that the analytic expression of the FTRL will be a softmax. But here I did not see the related analysis.

Minor point: At the end of the second paragraph of section 5, "it chooses the action with maximal composite risk in the decision making step.". It seems weird since the "risk" should be avoided instead of maximized. The authors should consider making it more clear.

**Q7 Justification For Your Score:**

Please refer to Q5.

**Q9 Complying With Reviewing Instructions:**

1: Yes.

---

### Official Review · Reviewer_PeBL · 2022-04-20

**Q2(1) Originality/Novelty:** 3
**Q2(2) Significance/Impact:** 3
**Q2(3) Correctness/Technical Quality:** 2
**Q2(6) Clarity Of Writing:** 3
**Q6 Overall Score:** 7
**Q8 Confidence In Your Score:** 3

**Q1 Summary And Contributions:**

The authors define a risk measure for risk-sensitive RL that combines aleatory and epistemic risk, rather than computing them individually or additively. They incorporate this measure within the distributional RL framework, and they use a bootstrapping framework to measure epistemic uncertainty (usually only aleatory is taken into account in DRL). This risk measure has a number of positive qualities, e.g. being coherent. It also resulted in fewer crashes in a simulated highway problem.

**Q2 Assessment Of The Paper:**

More detailed information regarding each of these aspects is given below:

**Q2(5) Reproducibility:**

3: Good: Key resources (e.g., proofs, code, data) are available and key details (e.g., proofs, experimental setup) are sufficiently well-described for competent researchers to confidently reproduce the main results.

**Q3 Main Strengths:**

* the risk measure maintains a number of properties that are theoretical, while also showing improved performance relative to other algorithms
* the paper is well written


**Q4 Main Weakness:**

* As far as I understand, the epistemic uncertainty is forced and controlled (because of splittingp the data) – is this unrealistic, and also only a fraction of the epistemic uncertainty?


**Q5 Detailed Comments To The Authors:**

n/a

**Q7 Justification For Your Score:**

The new risk measure appears to have both theoretical and practical benefits of previous ones.

**Q9 Complying With Reviewing Instructions:**

1: Yes.

---

### Decision · Program_Chairs · 2022-05-15

**Decision:**

Accept (Poster)

**Comment:**

Meta Review: The paper proposes a novel quantitative measure for risk in RL. The measure is sensitive to aleatoric and epistemic risk, and avoids some theoretical issues with previously proposed additive measures of the two types of risk. The paper also proposes a novel risk sensitive RL algorithm (based on ensemble bootstrapping and distributional RL). The method performs well empirically.

Pro:
* All reviewers agree that the proposed measure and algorithm is novel, sound, and original.
* 3 out of 4 reviewers agree that the paper is well written.
* The topic of risk sensitive RL is timely and important - 3 out of 4 reviewers rate the potential impact as good (3).
* Empirical results justify the theoretical claims and the method performs well compared to other distributional RL methods

Con:
* In their initial assessment, reviewers point out minor technical flaws and some difficulties in understanding parts of the empirical evaluation. The authors have commented on the issues raised, but only 1 reviewer has responded to the authors' comments.
* PeBL points out a potential shortcoming w.r.t. the training protocol (splitting the data). Authors acknowledge this and point to other papers in the literature using a similar protocol.
* The method seems somewhat complicated and could perhaps be presented better - this is based on questions raised by reviewers, and these questions should be helpful for the authors to improve the camera-ready presentation.

**Justification of recommendation:** 3 of 4 reviewers are in favor of accepting the paper. The only (weakly) negative reviewer has raised some major issues that are answered in the appendix of the paper (but the reviewer did not find the appendix) and answered in other sections of the paper (as pointed out by the authors). To me, the authors have adequately addressed all major and most minor comments in their extensive response (including the major issues raised by the negative reviewer) - unfortunately only the most positive reviewer responded to the rebuttal. Taking all this information together I can still confidently recommend acceptance of the paper.